# Mutual Information Based Learning Rate Decay for Stochastic Gradient Descent Training of Deep Neural Networks

**DOI:** 10.3390/e22050560

**Published:** 2020-05-17

**Authors:** Shrihari Vasudevan

**Affiliations:** IBM Research, Bangalore 560045, India; shrivasu@in.ibm.com

**Keywords:** deep neural networks, stochastic gradient descent, mutual information, adaptive learning rate

## Abstract

This paper demonstrates a novel approach to training deep neural networks using a Mutual Information (MI)-driven, decaying Learning Rate (LR), Stochastic Gradient Descent (SGD) algorithm. MI between the output of the neural network and true outcomes is used to adaptively set the LR for the network, in every epoch of the training cycle. This idea is extended to layer-wise setting of LR, as MI naturally provides a layer-wise performance metric. A LR range test determining the operating LR range is also proposed. Experiments compared this approach with popular alternatives such as gradient-based adaptive LR algorithms like Adam, RMSprop, and LARS. Competitive to better accuracy outcomes obtained in competitive to better time, demonstrate the feasibility of the metric and approach.

## 1. Introduction

Automated Machine Learning (AutoML) systems with Deep Neural Network (DNN) models are currently a very active research area [1] and key development goal being pursued by several major industry organizations, e.g., IBM, Google, Microsoft, etc. Among the key problems that need to be addressed towards this goal is hyperparameter selection and adaptation through the training process. Hyperparameter selection in DNNs is mostly done by experimentation for different data sets and models. In AutoML systems [1], this is realized through various forms of search including grid search, random search, Bayesian optimization, etc. Stochastic Gradient Descent (SGD) optimization [2] with mini-batches of data is a time-tested and efficient approach to optimizing the weights of a DNN. Hyperparameter selection and adaptation has a strong bearing on the outcomes of SGD-based training of DNN models. A key example of one such hyperparameter, which is also the subject of this paper, is the Learning Rate (LR). High LRs, particularly in early training stages, can result in instabilities and fluctuations in the parameter search process. Established procedures to set the LR to a low value at the beginning and then gradually warm up to the desired LR have been used effectively [3,4]. These approaches require the a priori definition of a policy or schedule and the LR changes according to this fixed policy. The fixed policy may not be suited for different data sets or model architectures which may be very different in complexity. Adaptive setting of the LR through the training cycle is one way of handling this issue. While adaptive LR algorithms based on gradients exist and are reviewed in the next section, there is incomplete understanding of the use of alternate metrics towards this objective and whether these could afford additional capabilities to the DNN training process. This paper explores the feasibility of using Mutual Information (MI) [5] as a metric to realize this objective.

## 2. Related Work

Adaptive learning rate (LR) schedules based on gradients have been proposed in various Gradient Descent (GD)-based optimization algorithms used for training deep neural networks; a survey of these is presented in [6]. These include AdaGrad [7], AdaDelta [8], RMSprop [9], Adam [10], and some more recent algorithms. Broadly, these set LRs at the level of individual parameters by considering the magnitude of past gradients; parameters associated with smaller past gradients are given a higher LR to enable larger updates as compared to those associated with larger past gradients. Depending on the data set and model complexity, careful initial selection of the LR may be required.

While adaptive gradient-based algorithms provide an excellent option for many scenarios, traditional (mini-batch) SGD is still the preferred option for situations involving complex models or data sets. Convergence properties of SGD have been studied in [2]; the paper demonstrates that subject to a few basic assumptions, an appropriately decreasing LR enables SGD to almost certainly converge to a minima. Various forms of decay in LR have been used, e.g., time-decay, step-decay, and exponential decay. Typically, in all of these cases, a decay-rate parameter and LR bounds (at least the maximum or starting LR) is required. The approach presented in this paper also requires the specification of LR bounds, a threshold based on change in Mutual Information (MI), and optionally, a starting LR. The difference arises in the nature of the decay and the new value of the LR. Established decay-LR SGD variants have a fixed rate of decay. The approach presented has a variable rate of decay governed by a threshold that is set to control search-space exploration at a given LR while favoring “exploitation” of the search-space as the LR decays. The new value of the LR set is based on a performance measure of the model. This work thus builds on the well established basis of (mini-batch) SGD using a decaying LR, but explores the viability of MI as a metric to automatically adapt the LR.

Recent works such as that in [11] use the training loss to adapt the LR for training the neural network model. The paper is based on linearizing the loss function at each epoch and finding its roots. At each epoch, the LR is set as the difference between current loss and the minimum achievable loss (observed thus far in the epoch) taken relative to the inner product between an estimated gradient and the update provided by a standard optimizer. The authors of [12] perform layer-wise adaptation of the LR to address the issue of large batch-size training of convolution networks. It observed significant variation in the ratio of the L2 norm of the weights of a layer to that of its gradients, between layers. Consequently, it proposed the LARS algorithm, which uses a network-level LR that decays exponentially with time (epochs). This global LR is further scaled locally for each layer using the ratio described before, computed for each layer. The approach presented in this paper explores the use of a layer-wise computable performance metric (MI) to adapt the LR, layer-wise, through the training cycle.

An automation of decaying LR SGD may be realized by different performance measures. Training accuracy is directly available and may be used; however, it does not provide a layer-wise performance measure for layer-wise LR setting, naturally obtainable using MI. Validation accuracy has the same issue but experience has also shown that it can be unreliable depending on the data set and partitioning. In principle, alternative performance measures, such as precision, recall, etc., may be used, but neither are these layer-wise metrics nor is the connection between a change in them and corresponding change in LR clear. MI provides a surrogate measure of classification accuracy [13]; in addition to capturing model performance, it can also be computed layer-wise.

Hu et al. [14] perform a study of information theoretic measures for objective evaluation of classifications. Several Normalized Information Measures derived from MI, divergence, and cross-entropy were considered. The paper suggests that measures from the first category were generally superior for data distinguishability. Meyen [15] links MI to classification accuracy through conditional (input-specific) classification accuracies. The work shows that MI and classification accuracy provide upper and lower bounds on each other through the conditional classification accuracy. It also suggests that MI and classification accuracy capture different aspects of the classification, that they are complementary and recommends that MI also be considered when developing classifiers. It is possible for two models to be identically accurate but for one to be more informative than the other; the latter may be expected to generalize better. This work explores an indirect approach to utilizing MI—the standard training pipeline (cost-function and optimizer) is not modified but the MI is used as a performance metric for adapting the LR through training.

Recent works of Tishby et al. [16] view deep neural networks through the Information Bottleneck (IB) [17] lens. In brief, IB theory is a MI-based signal compression–reconstruction formulation that attempts to find a maximally compressed signal abstraction that captures maximum information content of the signal. In the current context, the signal would correspond to the data being modeled (denote input as *X* and output as *Y*) and the abstraction would correspond to the layers of the neural network (denote as *T*). Shamir et al. [13] point out that the MI between a neural network layer and the input (MI(T,X) denoted here-on as ITX) functions as a regularization term and the MI between a neural network layer and the output (MI(T,Y) denoted here-on as ITY) functions as a measure of performance (e.g., classification accuracy). For a classification problem, the paper derives an upper bound on the misclassification error in terms of ITY. It also suggests that the amount of relevant information captured by the layer (or network up-to and including the layer) about *Y* is given by ITY/IXY, where IXY is the MI between the input *X* and output *Y*. Data Processing Inequality (DPI) [5] guarantees that ITY≤IXY for any layer or network.

Preliminary approach and experimental results of this paper were reported in [18]. This paper presents significant improvements over the previous version, significant change to the LR policy to enable it to build on the well understood convergence properties of decaying LR SGD and significantly better experimental results. The use of MI as a metric for LR adaptation is intended to lead to further work towards a deeper understanding of DNNs [19] and learning of DNNs through maximizing Mutual Information [16,20].

The contributions of this work are as follows.

A MI-based automation of decaying LR SGD training of neural network models that adaptively sets the LR layer-wise or for the whole network, through the training cycle.A LR Range Test that defines the broad LR bounds within which the proposed algorithm operates.Evaluation of the proposed algorithm in comparison with state-of-the-art alternatives applied to a range of data sets and models, to demonstrate the viability of the use of MI for automating decaying LR SGD training.

## 3. Approach

The proposed training algorithm is shown in Algorithm 1. It basically performs regular SGD-based Deep Neural Network (DNN) model training with an information driven LR setting every epoch. The model architecture and data set are first subject to an LR Range Test (LRRT) described in Algorithm 2; this yields broad LR bounds within which the algorithm operates $\left(LR_{min},LR_{max}\right)$, the LR $\left(LR_{top}\right)$ among the candidate LRs that produced maximum value of a metric (e.g., training accuracy) and a significance threshold ϵ. A small set of data is randomly sampled from the training data for MI computation. All MI computation measurement and reference upper bound are done only using this small subset only. Two metrics are computed during every epoch–$d_{p}$ is a measure of how far the model with current best MI is from the maximum value it can achieve and $d_{s}$ is a measure of the relative change in the MI between epochs. Both MI metrics used in the training algorithm are relative measures, enabling effective use of MI in the standard training pipeline without much computational overhead. Note that Algorithm 1 may be applied at the level of the network using the last layer for MI computation or may also be applied on a per-layer basis to set the LR of each layer independently. The value of $d_{p}$ decides the LR for the current epoch and the value of $d_{s}$ determines when the LR has to be changed.

The LR value is essentially set as $max(d_{p}\cdot \left(1-\frac{e}{E}\right)\cdot LR_{max},LR_{min})$. The first term reduces LR proportional to the current performance level relative to the maximum attainable; it implements a performance-based LR decay. The second term ensures LR reduction even in the face of the performance stagnating (e.g., a model that significantly under-fits the data). For models that are a priori known to fit the data well, this component can even be removed; experience suggests a small performance improvement may be obtained. The LR thus decays at a variable rate, with its value being defined by a performance component and a simple time-decay component. The LR is set as a function of $LR_{max}$ and is bounded by $\left(LR_{min},LR_{max}\right)$.
**Algorithm 1:** MI-based decaying LR SGD
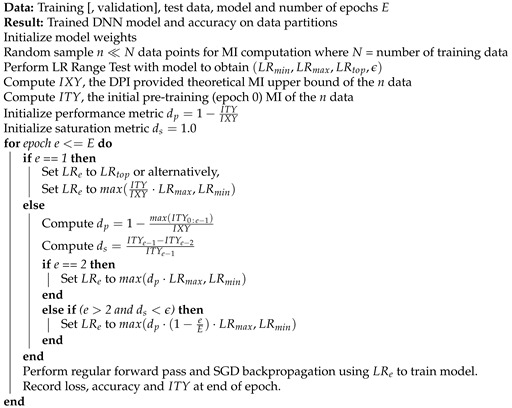


For the first epoch, the starting LR can be set to $LR_{top}$, obtained from the LRRT. It may also be set to a low value based on the initial pre-training MI as shown in Algorithm 1; therefore, the use of $LR_{top}$ is optional. Experience suggests both starting LRs perform well. Experiments reported in this paper have used $LR_{top}$. In the second epoch, the algorithm sets the LR based on the first epoch performance. Thereafter, LR changes occur when the change in MI metric $d_{s}$ is below ϵ; these changes incorporate maximum performance attained thus far and the simple time-decay component.

The state-of-the-art approach to initial LR determination by Smith [21] is based on a single run/model, a single metric, and continuous LR increase during the run. Although it is fast, it does not address repeatability of results. It cannot isolate the performance of a particular candidate LR, as performance depends on the previous LR and the initial model state. It cannot specify a layer-wise LR operating range in its current form and is essentially a manual process involving visual inspection, even if automation guidelines exist.

The LRRT procedure proposed in this work and described in Algorithm 2 essentially runs the training method described in Algorithm 1 for a fixed number of epochs, for a set of candidate LRs, and picks a LR range from their outcomes. The automated LR range selection process runs a multi-trial approach so as to enable repeatable outcomes. Each LR is tested from the same initial model state to enable fair testing, until training crashes and a new trial (model weights reset) begins. It records both standard training accuracy/loss metrics and also the MI measure ITY; the latter measure can also be recorded layer-wise to enable layer-wise specification of LR operating range. A crash in training is typically exemplified by a sudden or dramatic drop in all performance attributes. When multiple attributes are of interest (e.g., training accuracy and ITY), the LR bounds $\left(LR_{min},LR_{max}\right)$ can be computed for each attribute and the one with larger $LR_{max}$ can be chosen. In some cases, it may be possible to combine multiple attributes into a composite attribute and then extract the LR bounds as in the algorithm. Essentially, the LRRT picks LR bounds based on a fraction of the maximum performance; the first LR from which the growth exceeds this threshold and the first LR beyond which performance relative to maximum performance drops below this threshold constitute the required LR bounds. The process is automated and results in $\left(LR_{min},LR_{max},LR_{top},\epsilon \right)$. For experiments in this paper, the bound selection uses training accuracy and the significance threshold ϵ is based on the MI saturation metric, as required of the training algorithm.
**Algorithm 2:** LR Range Test
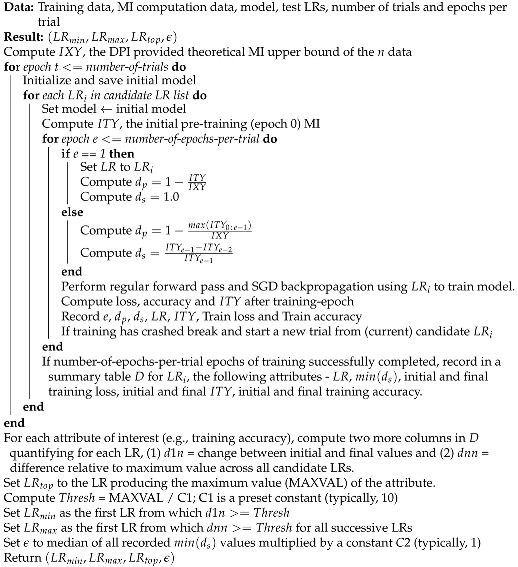


When the LRRT is performed in time constrained scenarios (e.g., AutoML systems running on time budgets), fewer trials and epochs per trial (e.g., two trials of three epochs each) may be used. While the bounds will remain largely unaffected, the significance threshold ϵ may be larger leading to frequent LR decay that would resemble a continuous time-decay curve. This effect may be offset by setting constant C2<<1 in Algorithm 2. Extensive experimentation suggests that budget permitting, a LRRT of at least three trials of five-to-six epochs each would be useful. A small ϵ would lead to relatively less frequent LR decay; the resulting LR plot would resemble a step-decay curve.

Bottou [2] demonstrated the almost certain convergence property of SGD subject to the assumptions on (1) differentiabilty of the cost function, (2) two conditions on the learning rate (LR), (3) constraint that moments of the update do not grow too quickly relative to the change in parameters, and (4) constraint that the cost function cannot have a plateau region within which the parameters can grow indefinitely without ever being able to leave. The two conditions on the LR require that it decreases rapidly but not so quickly so as to prevent the algorithm from reaching the minima. These are respectively quantified as $\sum_{i=1}^{\infty }\gamma_{i}^{2}<\infty \;\mathrm{and}\;\sum_{i=1}^{\infty }\gamma_{i}=\infty$, where $\gamma_{i}$ are the LR values. The approach presented in this paper automates the LR decay using a performance-based variable-rate decay of the LR, to enable both layer-wise and full-network LR setting; the paper explores the use of a MI-based performance metric in this context. The approach may thus leverage established ideas on SGD convergence. Using a fixed decay could result in unnecessary dwelling at a relatively high LR even when the model has quickly trained to a competitive performance outcome. This would lead to slower convergence. It is important to note that no change to the loss function is pursued in this work. No change to the standard SGD training pipeline is proposed. Consequently, the proposed approach also does not affect the overfitting properties of the standard DNN training pipeline. The proposed approach replaces a fixed parameter-based decaying LR SGD with a variable performance-based decaying LR SGD; automated approaches to parameter selection and a layer-wise performance metric are also proposed.

## 4. Experiments

Experiments were conducted on multiple standard image classification data sets with different standard DNN architectures; they compared the proposed approach with one or more gradient-based adaptive LR algorithms such as Adam [10] and RMSprop [9], standard step-decay SGD, and the weight-norm/gradient-based layer-wise adaptive LR algorithm (LARS) [12]. The data sets used include MNIST [22], CIFAR-10, CIFAR-100 [23], and Imagenet-1K [24]. The CIFAR-10 data set was tested with two different architectures—the AllConvNet [25] and the VGG-16 [26]. Layer-wise testing of the proposed approach was limited to MNIST and CIFAR-10 applied to the AllConvNet. In all experiments, the best test accuracy of 3 random seed runs is reported; unless specified, the deviation in outcomes between the 3 runs was within 0.5%. Each run trains a model from scratch. All models use ReLU activations for all layers except the last one which uses a Softmax. SGD training used a fixed momentum of 0.9. Nesterov acceleration was used for training MNIST and CIFAR-10 (AllConvNet); for CIFAR-10 (VGG-16), CIFAR-100 and Imagenet-1K, standard settings including weight-decay were used. For each data set, a LRRT was performed to determine the LR operating range before applying the proposed algorithm. Layer-wise testing of the proposed approach applied the exact same LR operating range for each layer. It is also possible to conduct a layer-wise LRRT to determine layer-wise LR operating ranges. For all alternative approaches (Adam, RMSprop and LARS), a first test was performed to determine the best initial LR from a range of options (about 10). This was followed by 3 tests with this LR; the results report the best of these outcomes.

Computing MI is computationally expensive; performing MI computation after each epoch in deep neural network training can prove to be infeasible. This paper uses the Kraskov–Stögbauer–Grassberger (KSG) estimator [27] for MI estimation; other algorithms could also be used. The KSG estimator is approximate in that it adds a small jitter (∼$e^{-10}$) to overcome data degeneracies. This paper relies on two ideas to effectively use MI in training with large data sets: (1) use a subset of data for MI computation; plotting the MI vs sample size curve for different data sets enables informed selection of an appropriate subset sample size for per-epoch MI computation, and (2) the approximate MI value may suffice if the relative measures can be utilized for the problem. Based on Figure 1, a sample size of 1000 was chosen for MI computation in the experiments of this paper; this number was a trade-off between computational overhead (due to the chosen size) and variation in the MI estimate. The Experiments section also describes a sample size sensitivity test to check the practical impact of the selected size on training outcome of training. A large sample size while providing a more accurate MI estimate would add significant computational overhead to the training process. Since accurate MI estimation is not a required objective of the approach presented in this paper, the approach is developed on an approximate estimate of the MI.

MNIST: Data comprises 60,000 training and 10,000 testing grayscale images of size 28 × 28, representing 10 class outcomes. A model (see Figure 2) based on the LeNet-5 architecture [22] was trained on the data. The model had two sets of convolution and pooling blocks followed by 2 fully connected layers separated by a Dropout layer. Training was done for 50 epochs using a batch size of 256. No data transformation other than basic scaling of data to [0,1] was performed. Adam and RMSprop each reported a best accuracy of 99.53%. The proposed approach reported a best outcome of 99.27% when trained using a single LR (see Figure 3) and 99.39% when trained using layer-wise LR based on its MI (see Figure 4). The LARS algorithm reported a best accuracy of 99.43%. Figure 4 suggests layer-wise training may be beneficial because the earlier layers continue tuning the model weights for much longer, resulting in slower decay of LR, whereas the last layers mapping abstract features to outcomes reaches its desired state quickly, resulting in rapid decay of LR.

CIFAR10: Data comprises 50,000 training and 10,000 test color images of size 32 × 32, representing 10 class outcomes. The data was first trained with a model based on the AllConvNet architecture proposed in [25]. The model implementation used the All-CNN-C architecture from the paper. However, no data augmentation was used. The model implementation used dropout (50%) only after max-pooling layers, no L2 regularization for weights, a fixed momentum value of 0.9, Nesterov acceleration, and a batch size of 256; these choices were made based on preliminary tests. Moreover, these tests suggested that setting C2 = 1 leads to a very quick LR decay; so for this data set, C2 was set to 0.01 (experimentally determined) to determine LRRT parameters. As in the cited paper, training was done for 350 epochs. Adam and RMSprop reported best accuracies of 87.84% and 88.13%, respectively. The proposed approach reached 88.86% when trained with a single LR for the entire model (see Figure 5) and 87.77% when trained layer-wise (see Figure 6). The deviation between the best and worst outcomes of the proposed approach was just under 1%. LARS reported a best outcome of 77.03% if weight-decay was not used, to enable a fair comparison with proposed and other approaches. If weight decay was included, LARS could obtain a best outcome of 89.91% across 3 runs. In comparison, the best layer-wise outcome of the proposed approach, with further manual tuning of parameters but without weight decay, was 88.79%.

Experiments until now used Keras/Tensorflow implementations of different models; those that follow are based of standard PyTorch model implementations. As a next step, CIFAR10 data was also subject to training and evaluation with the VGG-16 architecture. Standard training choices of a batch-size of 256, weight-decay of $5\times 10^{-4}$, momentum of 0.9 and basic data transformations through random-crops and random horizontal flips were used. Training was performed for 200 epochs with a batch size of 256. Preliminary tests suggested that C2 needed to be set to 0.005, to prevent an overly quick LR decay. The outcome of the LRRT was then used to train the model using the proposed approach. Adam reached a best accuracy of 89.62%, while RMSprop could reach a best accuracy of 87.17%. The proposed approach reached 92.21%. A plot of the accuracy and the LR is shown in Figure 7. A step-decay-LR SGD approach that decayed the LR by 2 every 30 epochs was found to be the best alternative, reaching 92.58%. The proposed MI-based SGD approach was very similar to the step-decay-LR SGD in terms of speed, with the former reaching 91% and 92% at epochs 65 and 102, respectively, while the latter reached these milestones at epochs 64 and 97, respectively.

Experiments in this paper use a sample size of 1000 data samples to estimate MI during every epoch of the training process. To test the sensitivity of the accuracy outcomes on the sample size of the data used for MI computation, the CIFAR10 data with VGG16 model was tested with two other sample sizes: half of that used for experiments in this paper (500) and double of that used for the experiments in this paper (2000). As with all other experiments in this paper, all numbers reported here are best outcomes of three random seed runs. First, the previously estimated parameters (from the experiment above) were applied in both cases. This would be indicative of the sensitivity of the accuracy outcome on Algorithm 1 with a fixed parameter set but different MI sample sizes. Using a sample size of 500 produced a best test accuracy of 92.37% and using a sample size of 2000 resulted in a best test accuracy of 92.26%. The maximum variability in test accuracies between runs, in both cases, was under 1.7%.

Next, the parameters used were re-estimated using the different sample sizes (with the same value of C2 used above) and the accuracy outcomes were tested again. This would primarily indicate the sensitivity of Algorithm 2 on MI sample size, but also performs a second sensitivity evaluation of Algorithm 1, with the new set of parameters found. Using both a sample size of 500 and 2000, near identical parameter outcomes for each of $\left(LR_{min}\;=\;0.0003,\;LR_{max}\;=\;0.06,\;LR_{top}\;=\;0.02,\;\epsilon \;=\;0.00054\right)$ were obtained; these were almost identical to those obtained earlier using a sample size of 1000 $\left(LR_{min}\;=\;0.0003,\;LR_{max}\;=\;0.07,\;LR_{top}\;=\;0.01,\;\epsilon \;=\;0.00048\right)$. Subsequent execution of Algorithm 1, with the new parameters, produced a best test accuracy of 92.76% with a sample size of 500 and 92.23% with a sample size of 2000, with an inter-run variability of ~1% in both cases. The best accuracy of 92.76% is marginally higher than the best outcome obtained across all other approaches. These outcomes suggest that (a) using a small sample size for MI computation does not affect performance of the proposed approach; it would make computational overhead due to MI computation negligible, and (b) the proposed approach is relatively robust to the sample size chosen for per-epoch MI computation. It is likely that the use of a small but representative sample of the training data along with the use of relative MI metrics affords this capacity to the proposed approach. It must be noted in this context that accurate MI estimation is not the goal of this paper; efficient automation of the training process, using MI, is realized by the proposed approach.

CIFAR100: Data comprises 50,000 training and 10,000 testing color images of size 32 × 32, representing 100 class outcomes. Experiments were conducted using the Wide-Resnet-28-10 architecture, proposed in [28]. Training was done for 200 epochs using a batch size of 128. Adam reported a best case performance of 75.02% while RMSprop performed comparably at 74.74%. The proposed approach reached 81.25%. The author’s code (baseline) and fixed LR-decay policy reached 81.76%. While the proposed approach was competitive, it was also faster in that it reached 80% and 81% accuracy in 93 and 107 epochs respectively compared to the fixed LR policy used by the author which took 121 and 130 epochs respectively to reach the same levels. Plots of the accuracy and the LR over training are shown in Figure 8.

Imagenet-1K: Data comprises 1.2 Million training and 50,000 test color images, representing 1000 class outcomes. Experiments were conducted using the Resnet-50 model architecture proposed in [29]. Training was done for 100 epochs using a batch size of 256. Standard data transformations through resizing to desired size, random-crops, and random horizontal flips were used. Preliminary tests observed the MI growth over a few epochs and when compared to the desired theoretical upper bound, determined that using a value of 30 for C1 in the LRRT would be appropriate. For this data set, the proposed approach was compared with Adam and a widely used fixed LR-decay policy involving starting from a LR of 0.1 and stepping-down the LR by a factor of 0.1 every 30 epochs. Adam reported a best test accuracy of 69.82% only. The fixed LR-decay policy reached 75.57%. The proposed approach reached 76.05% for the same extent of training, which is competitive with state-of-the-art outcomes [4] on this data set. Plots of the accuracy and the LR over training are shown in Figure 9.

## 5. Discussion

Algorithm 2 presented an approach to automatic selection of hyperparameters for training the deep neural network model using Algorithm 1. The default values of C1 and C2 in Algorithm 2 generally work well across models and data sets. In cases where the data set is large or complex (e.g., characterized by slow accuracy growth) or the number of epochs is small or very large, C1 and C2 may need to be set to different values, to obtain better outcomes than alternative approaches. In this paper, this was done empirically. The experiments in this paper suggest that Algorithm 2 needs further understanding in terms of automatic setting of C1 and C2 while considering the number of epochs of training available and data set complexity, to enable it to be fully automated in all possible application scenarios, while also producing the best outcomes. This requires separate exploration and is intended to be pursued as a future extension of this work.

Experiments of this paper demonstrated the need for tuning of hyperparameters for every approach, including the state of the art in adaptive LR algorithms, e.g., Adam and RMSprop. Algorithm 1 presented an MI-based automated training approach for deep neural networks. It required the definition of three parameters: the LR bounds and a significance threshold to control the decay of the LR (the fourth parameter, $LR_{top}$, is optional). These were set by performing a LR range test, presented in Algorithm 2. The definition of bounds instead of a single starting LR alone was done to explore dynamic LR adaptation (both increase and decrease of LR during training) in response to changes to other hyperparameters (e.g., batch-size) or other factors (e.g., computational resources). A preliminary foray in this direction was documented in [18], but much more exploration is intended as part of future work. Algorithm 2 also has a couple of parameters (C1 and C2) that control the actual selection of the LR bounds within which Algorithm 1 operates. Considering this aspect, unlike alternative training approaches, the proposed approach pushes parameter selection one level above in the hierarchy, i.e., it uses (default or other coarse) values for C1 and C2 that in turn do hyperparameter bound selection (done by Algorithm 2) for the actual training algorithm (Algorithm 1). It is possible that this meta-level parameter selection strategy will enable reduced sensitivity of outcomes on the values selected; this hypothesis will be verified alongside the proposed future extension stated above.

This paper explores the use MI-based metrics to automate the LR decay in SGD training of deep neural networks. The experiments compare the proposed approach to state of the art adaptive LR methods as well as widely used fixed LR decay policies for SGD. In comparison with the fixed LR policies, the proposed approach is competitive or better in terms of accuracy or convergence speed because it prevents unnecessary dwelling or insufficient exploration of the search space by using a performance-based metric over a manually set policy. In terms of the specific performance metric, the use of MI-based performance metrics, at the minimum, affords a layer-wise training capability over the use of a metric such as the training accuracy. The proposed approach to SGD training of neural networks perform favorably in comparison with state-of-the-art gradient-based adaptive LR approaches like Adam, RMSprop, etc. Recent works have attempted to explain this finding [30] and also exploit it [31] in training deep neural network models. The former paper found that for problems where the number of parameters exceeds the number of data points, adaptive gradient methods generalize poorly compared to SGD, even if the training performance suggests otherwise. The paper attributes this to a tendency of adaptive gradient methods to overfit features that happen to have no value towards generalization. They also point out that adaptive gradient-based methods require similar amounts of hyperparameter tuning as compared to other approaches; this was observed in the experiments of this paper as well.

While experiments in this paper were developed and tested in an image classification application context, an experiment was conducted to understand if the proposed approach would also work in a regression context. A recently collated temperature prediction data set [32] was used to test if the proposed approach could train neural network models for regression problems. The paper uses two identification attributes (station and time), fourteen numerical weather prediction model attributes, two in situ temperature observations and five geographical attributes to forecast maximum and minimum next-day temperatures. A total of 7750 data with 25 attributes were provided; the data spanned the years 2013 to 2017. For the purposes of this experiment, data from 2013 to 2016 (80% of the entire data set) was used to train a neural network model, which was used to predict the minimum and maximum temperatures for each data instance in 2017 (20% of the data set). A simple neural network comprising 3 fully connected layers, with 64, 32, and 2 neurons, respectively, was used to perform regression. All neurons used ReLU activations. As with other experiments in this paper, MI estimation per training epoch was done using 1000 data points. As a one time computation on real data, the maximum possible MI was estimated using the entire data set. The model was trained using Adam, RMSprop and the proposed approach, for 2000 epochs. Given the context of a regression problem, the Mean Squared Error was used as a loss metric to perform the optimization and the results report the Mean Absolute Error (MAE) (temperature in degrees Centigrade) in prediction of both minimum and maximum temperature, over the test data subset. Adam and RMSprop were subject to an LR search process to find the best starting LR. As an accuracy measure is not available, the LRRT parameter selection was done using the MI metric, logged simultaneously, demonstrating the flexibility of the LRRT algorithm. The proposed approach followed the steps of the paper in performing an LR Range Test (Algorithm 2) followed by the MI-based SGD training (Algorithm 1). Adam reported a best MAE of 0.97 ${}^{{}^{\circ}}$C over three random seed runs while RMSprop reported a best MAE of 1 ${}^{{}^{\circ}}$C. The proposed approach reached a competitive best MAE of 1.32 ${}^{{}^{\circ}}$C over three random seed runs; its outcome is depicted in Figure 10. The deviation between individual outcomes was under 0.03 ${}^{{}^{\circ}}$C in all cases. For the proposed approach, the LR Range Test used C2 = 0.01, as was used for CIFAR-10. An aspect that needs further understanding is that the proposed approach was significantly slower than Adam in this experiment; the proposed approach first reached a MAE between 1.3 and 1.4 just after 1160 epochs, whereas Adam had attained a minimum MAE of just under 1 ${}^{{}^{\circ}}$C in the same time. Although the experiment demonstrates of the approach working in a regression problem, further experiments on other data sets and models are required to conclusively quantify the performance of the proposed approach in regression problems and derive insights on the contexts (e.g., model/data complexity) where existing adaptive LR approaches and the proposed MI-based training approach are likely to perform best.

## 6. Conclusions

The paper proposed a novel Mutual Information (MI)-driven, decaying Learning Rate (LR) Stochastic (mini-batch) Gradient Descent (SGD) training approach for Deep Neural Network models. The paper also introduced a novel multimetric LR Range Test to automatically select LR bounds and decay rate parameters for the training algorithm being proposed. Experiments reported demonstrate that the proposed approach produced competitive to better outcomes across data sets, neural network architectures, problem contexts (classification or regression), and training choices. The experiments also demonstrated the ability of using MI for both regular model-wise training and layer-wise training. A more efficient implementation of the MI computation algorithm using GPUs and suitable data structures may enable the application of this approach to wider/deeper neural network models. Overall, the paper demonstrated that MI can be used as a metric for performance-based decaying LR SGD training, leading to competitive outcomes compared to the best alternatives for a range of data sets and models. 

## Figures and Tables

**Figure 1 entropy-22-00560-f001:**
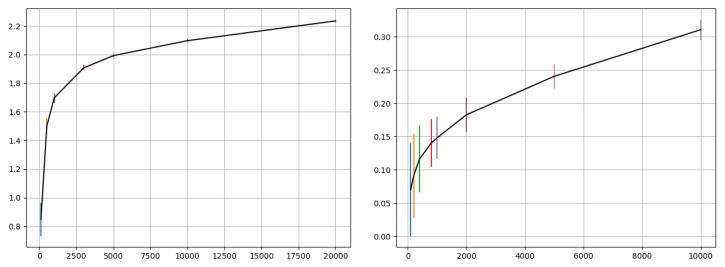
Mutual Information (MI) (of input and output training data) vs. sample size for MNIST (**left**) and CIFAR-10 (**right**) as computed using the KSG estimator. The plots show estimated mean and standard deviation (error bar) for each sample size tested. A sample size of 1000 was chosen for MI computation in the experiments of this paper—this was selected as a trade-off between computational cost of computing MI and the variation in estimates. A sample size sensitivity test using CIFAR-10 is described in the experiments.

**Figure 2 entropy-22-00560-f002:**
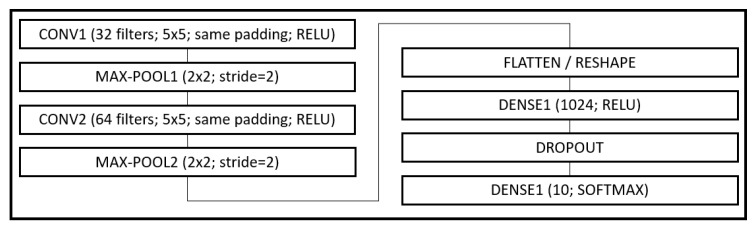
MNIST model description.

**Figure 3 entropy-22-00560-f003:**
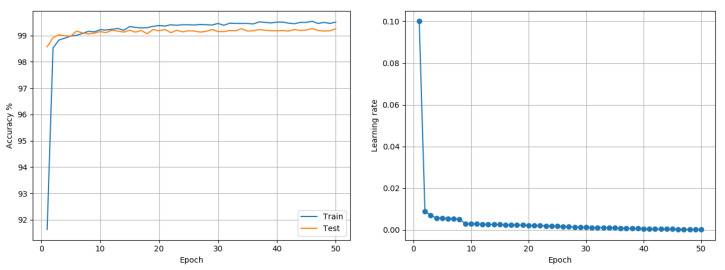
MNIST (Model based on LeNet-5): Accuracy and LR plots using the proposed approach and single model-level LR in [0.0001, 0.2]. The proposed approach produced a best test accuracy of 99.27% in 50 epochs, compared to the best alternative of 99.53% obtained using both Adam and RMSprop. Best outcomes from 3 random seed runs are reported.

**Figure 4 entropy-22-00560-f004:**
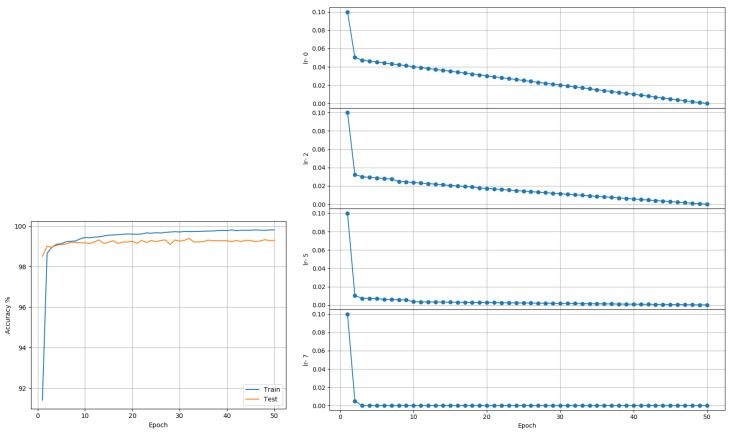
MNIST (Model based on LeNet-5): Accuracy and LR plots using the proposed approach and layer-wise LR in [0.0001, 0.2]. The proposed approach produced a best test accuracy of 99.39% in 50 epochs, compared to the best alternative of 99.43% obtained using LARS. Best outcomes from 3 random seed runs are reported.

**Figure 5 entropy-22-00560-f005:**
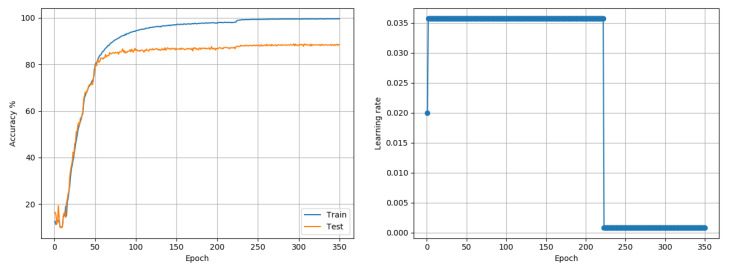
CIFAR10 (AllConvNet): Accuracy and LR plots using the proposed approach and a single model-level LR in [0.00075, 0.04]. The proposed approach produced a best test accuracy of 88.86% in 350 epochs, compared to the best alternative of 88.13% obtained using RMSprop. Best outcomes from 3 random seed runs are reported.

**Figure 6 entropy-22-00560-f006:**
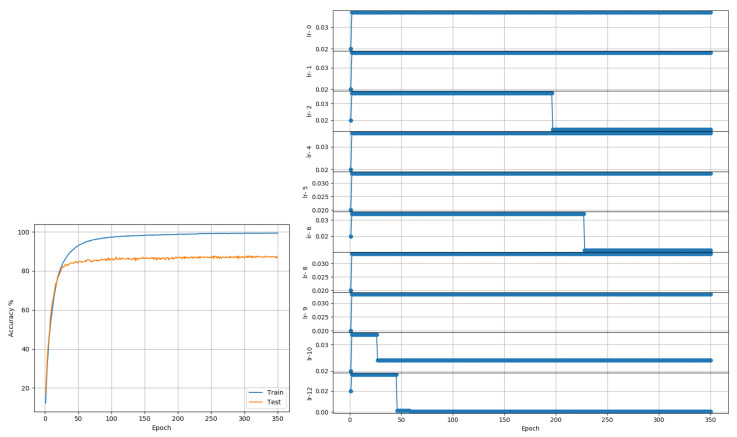
CIFAR10 (AllConvNet): Accuracy and LR plots using the proposed approach and layer-wise LR in [0.00075, 0.04]. The proposed approach produced a best test accuracy of 87.77% in 350 epochs, compared to the best alternative of 77.03% obtained using LARS. Best outcomes from 3 random seed runs are reported.

**Figure 7 entropy-22-00560-f007:**
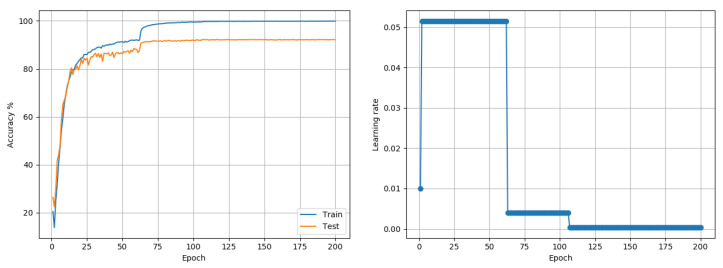
CIFAR10 (VGG16): Accuracy and LR plots using the proposed approach and and a single model-level LR in [0.0003, 0.07]. The proposed approach produced a best test accuracy of 92.21% in 200 epochs, compared to the best alternative of 92.58% obtained using SGD with fixed LR decay policy. Best outcomes from 3 random seed runs are reported.

**Figure 8 entropy-22-00560-f008:**
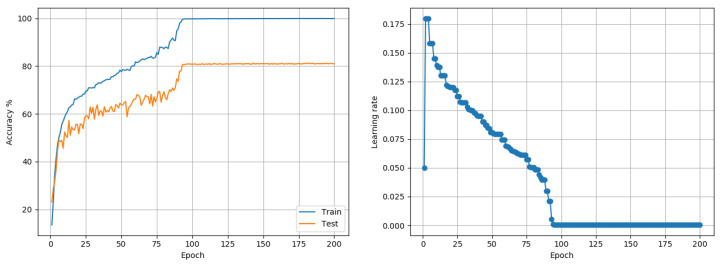
CIFAR100 (Wide-Resnet-28-10): Accuracy and LR plots using the proposed approach and a single model-level LR in [0.0003, 0.07]. The proposed approach produced a best test accuracy of 81.25% in 200 epochs, compared to the best alternative of 81.76% obtained using SGD with a fixed LR decay policy. The proposed approach reached top-level accuracies 10–15% faster than the alternative. Best outcomes from 3 random seed runs are reported.

**Figure 9 entropy-22-00560-f009:**
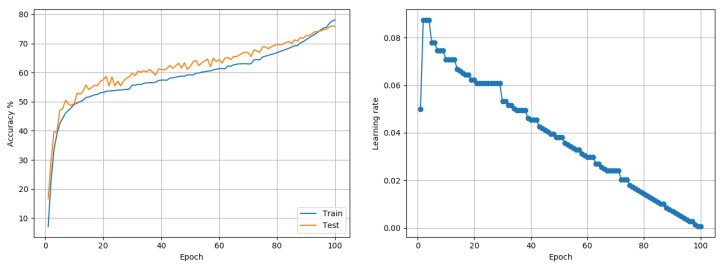
Imagenet-1K (Resnet-50): Accuracy and LR plots using the proposed approach and a single model-level LR in [0.0005, 0.1]. The proposed approach produced a best test accuracy of 76.05% in 100 epochs, compared to the best alternative of 75.57% obtained using SGD with a fixed LR decay policy. Best outcomes from 3 random seed runs are reported.

**Figure 10 entropy-22-00560-f010:**
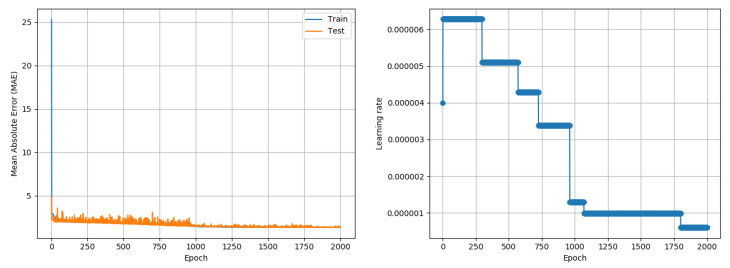
Results of the application of the proposed approach to a temperature prediction (regression problem) data set [32]. The proposed approach produced a competitive Mean Absolute Error (MAE) of 1.32 ${}^{{}^{\circ}}$C in comparison to the best alternative approach (Adam) which produced an MAE of 0.97 ${}^{{}^{\circ}}$C. Reported numbers are best outcomes of three random seed runs.

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
