# Peer review of "Mutual Information Based Learning Rate Decay for Stochastic Gradient Descent Training of Deep Neural Networks"

_entropy, 2020, doi:10.3390/e22050560_

Round 1

Reviewer 1 Report

This paper presents a new method for learning rate decay in SGD algorithm for neural network training. It is well written and technically sound. Although there have been some studies trying to use the MI criterion directly in the training process, this study shows how to tie the MI with the LR in practice. The experiments are well described and the results suggest that the presented method performs on par with some well-established methods such as Adam and RMSprop.

There are, however, some issues that need to be cleared before the publication:

  1. The proposed method is tested only on the classification task. If it works for regression, then it should be tested on a regression task as well. If not, this should be clearly stated in the abstract and the body of the paper.
  2. Both of the proposed algorithms work on a small set of randomly sampled data from the training set. This, naturally, raises the question "How the amount of sampled data (n in the paper) affects the algorithms results and in turn the NN training and performance?". There are no numbers for n given nor a suggestion for how to choose it. This is an important hyperparameter of the proposed method and needs experimental evaluation.
  3. There are two more important hyperparameters - C1 and C2 and their values according to the experiments are quite different from the "typical" ones given in Algorithm 2. This means that NN training performance is quite sensitive with respect to their values. Again, this needs to be shown experimentally. 
  4. The author claims that in some cases, the proposed method achieves the best NN classification performance faster than the other methods. This may be true but considering the time needed for running Alg.1 and 2, the total NN training time may be longer. This also needs experimental evidence.
  5. The proposed algorithm has several hyperparameters and so do the others. In this respect, a comparison in terms of the number of hyperparameters and the performance sensitivity with respect to their values is necessary.

Some small comments:

  1. Line 54: The word "metric" is written two times.
  2. Line 116: The "data" apparently refers to the MI computation data, but they are defined later in the text.
  3. Line 142: "is based on based" -> "is based"
  4. Figures 2-8 are too big.

Reviewer 2 Report

In the paper, the author described a mutual information for the neural networks training process. In general, the idea is interesting but the paper is very simple and does not provide basic information about the idea like a mathematical model, or proofs for it, no experimental results on complex datasets and simple comparison with other existing solutions. I must reject this paper because of the poor described idea. My main issues that should be a concern by the author:

1) The abstract does not provide any information on what is the novelty, why it is better than other algorithms.
2) The introduction section should analyze the problem of deep architecture and presents current development in this area from the last 3-4 years. In this time, the heuristic approach for training CNN architecture was used and the obtained results are promising. In the reviewed paper, there is no information and no analysis of the latest knowledge.
3) Section cannot be named as 'Approach' - it does not provide any information for readers. The first section is like a related works section, not the presented approach.
4) In section 3.2, the authors use some unknown operator like star *, if it was multiplication, there should be \cdot.
4) In Section 3.2, there are some words about the proposed idea, but the mathematical model is completely missing.
5) Where is a mathematical proof for this proposal?
6) Experiments were carried out on simple datasets like MNIST and CIFAR, which are basic datasets and for showing results, there should be some more complex datasets used.
7) Authors show some results, but there is no deep discussion on this, no comparison with other techniques.
8) Did you try to use freezing layers?
9) What about other classifiers than learning transfer?
10) What about other neural architecture like belief, etc?
11) Where is the statistical analysis of the results?
12) Where is a comparison with heuristic/BP/Adam/Ada on different architectures?
13) How this coefficient was chosen?
14) Conclusions are not supported by results, because of no mathematical/theoretical analysis, no complex experiments.

Reviewer 3 Report

This manuscript introduces an algorithm for deep neural networks training. Specifically, it employs the concept of mutual information between the neural network output values and the corresponding true values to optimize the learning rate in training stage. The algorithm can be nested under the stochastic gradient descent concept.

The algorithms can be used per layer or on the whole network and it competes with several other options, to which it claims better results within a selected range of datasets.

Beyond algorithms, the design of the research underlying this manuscript also makes the case for mutual information as a metric for training performance within the range herein tested.

I do not believe this manuscript title accurately describes the research. Not only can “Information driven” encapsulate very different methods and have very different meanings for each reader but also it is not clear about the manuscript content. Furthermore, acronyms, straightforward can it be, shall not be employed without explanation. That is certainly not the case for a manuscript title.

The theme is current and interesting, even if the Author did not a great job in exposing the research gap magnitude with such a superficial definition.

It is true that “Hyper-parameter selection in deep neural networks is mostly done by experimentation for different data-sets and models. A key example of one such hyper-parameter that is the subject of this paper is the learning rate” Yet, most researchers do not regard this as a critical hindrance, since an experimented operator is able to effectively attain adequate parameters with minimal experimental effort. Furthermore there are competing approaches which were not stated in the problem definition.

While a critical assessment of past research was offered (not only in the due section but originally along most of the manuscript) – which nowadays is rare and must be prised – it stands out that related work research was very scarce. Much more could, and should, have been done.

Structurally, first and second sections do not make much sense. The introduction is exiguous and does not successfully introduce the theme, the problem, the motivation and the work. Yet, part of those has been nested under section two, including the work contributions.

State-of-the-art, as a critical analysis of past work, is well done, but upon a very small references list. This leaves unaddressed most of the current main references and latest developments in the field.

Further on, third (approach) section returns to related work analysis.

Accounting, also, for the fact that redaction is not a strong point, this makes the manuscript unnecessarily difficult to understand, which is unfortunate since it bears a very good, but dispersed critical analysis.

Therefore, I believe that a complete reorganization, expansion and enhancement should be performed for sections 1, 2 and part of 3.

There are not unjustified self-citations.

Research methods are adequate and sufficiently explained.

The research beneath the manuscript is substantial and replicable to a certain degree.

While this type of practical research is much more directed for speciality journals, I believe it can be fitted into Entropy journal.

Conclusions are significant and supported by the attained results. However, results discussion is far from deep.

Regarding the research novelty, I believe that the Author offered an interesting development. This, obviously, cannot be credited as a major breakthrough if one attends to the existing work in this field, as well as the fact that networks training can be effectively achieved without it, but it is an innovative work.

The document redaction in general and, particularly, sentences organization, is not a strong point. Yet, it is intelligible. Thus, I recommend an extensive writing revision.

All things considered, I believe that this manuscript is a good scientific contribute and shall be published, but some improvements and further developments are needed, following the aforementioned comments.

The research strongest point is, from my point of view, offering a practical advance for deep neural networks training, which can easily applicable and validated given this work reproducibility. The results are good, even not “revolutionary” and deserve attention from the scientific community.

The former clearly overtake the manuscript most notorious weaknesses, which include the redaction, organization and exiguous introduction and theme framing as well as insufficient results discussion.

Round 2

Reviewer 1 Report

The revised paper addresses most of the mine concerns and I think it is suitable for publication.

Reviewer 2 Report

Accept